# Experience with Optical Coherence Tomography Enhanced by a Novel Software (Ultreon™ 1.0 Software)—The First One Hundred Cases

**DOI:** 10.3390/medicina58091227

**Published:** 2022-09-05

**Authors:** Stanisław Bartuś, Wojciech Siłka, Karol Kasprzycki, Karol Sabatowski, Krzysztof Piotr Malinowski, Łukasz Rzeszutko, Michał Chyrchel, Leszek Bryniarski, Andrzej Surdacki, Krzysztof Bartuś, Rafał Januszek

**Affiliations:** 1Department of Cardiology and Cardiovascular Interventions, University Hospital, 30-688 Kraków, Poland; 2Institute of Cardiology, Jagiellonian University Medical College, 31-202 Kraków, Poland; 3Students’ Scientific Group, the Second Department of Cardiology, Jagiellonian University Medical College, 31-034 Kraków, Poland; 4Faculty of Medicine, Department of Bioinformatics and Telemedicine, Jagiellonian University Medical College, 31-530 Kraków, Poland; 5Center for Digital Medicine and Robotics, Jagiellonian University Medical College, 31-530 Kraków, Poland

**Keywords:** clinical outcomes, optical coherence tomography, percutaneous coronary intervention, procedural indices, stent expansion

## Abstract

*Introduction:* Optical coherence tomography (OCT) intravascular imaging including the latest version Ultreon™ 1.0 Software (Abbott Vascular, Santa Clara, CA, USA), not only improve patients prognosis, but also facilitates improved percutaneous coronary intervention (PCI). *Objectives:* The aim of the study was to compare procedure related decision making, procedural indices, clinical outcomes according to the extent of stent expansion and assess risk factors of underexpansion in patients treated with PCI using OCT. *Methods:* The study comprised 100 patients, which were divided in groups according to the extent of stent expansion: <90 (29 patients) and ≥90% (71 patients). Comparison of OCT parameters, selected clinical and procedural characteristics was performed between groups. We assessed clinical outcomes during the follow-up: major adverse cardiovascular events and risk factors of stent underexpansion. *Results:* Patients from the stent underexpansion group were treated more often in the past with percutaneous peripheral interventions (*p* = 0.02), no other significant differences being noted in general characteristics, procedural characteristics or clinical outcomes comparing both groups. Significant predictors of stent underexpansion assessed by simple linear univariable analysis included: hypercholesterolemia, obstructive bronchial diseases and treatment with inhalators, family history of cardiovascular disease, PCI of other than the left main coronary artery, stent and drug-eluting stent implantation, PCI without drug-eluting balloon, paclitaxel antimitotic agent, greater maximal stent diameter and lower mean Euroscore II value. Univariable logistic regression analysis revealed a correlation between stent underexpansion and greater creatinine serum concentration before [OR: 0.97, 95%CI: 0.95–0.99, *p* = 0.01)] and after PCI [OR: 0.98, 95%CI: 0.96–0.99, *p* = 0.02)]. *Conclusions:* Based on the presented analysis, the degree of stent expansion is not related to the selected procedural, OCT imaging indices and clinical outcomes. Logistic regression analysis confirmed such a relationship for creatinine level.

## 1. Introduction

One of the main purposes of using the intravascular imaging technique during percutaneous coronary interventions (PCI) is to improve procedural and clinical outcomes. Among key factors responsible for establishing long-term clinical outcomes expressed as revascularization due to in-stent restenosis is stent underexpansion. The adoption of optical coherence tomography (OCT) in clinical practice has been reported in several studies and improved outcomes were demonstrated compared to angiography alone [1,2]. In previous research, it has been demonstrated that pre- and post-PCI OCT had significant effects on changing decisions in treatment strategy [3,4,5]. Procedural success in patients treated with PCI assisted by OCT has been defined as the percentage of patients achieving optimal (>95%) and acceptable (90% to 95%) stent expansion, where minimum stent expansion is defined as a minimum stent area of at least 90% in both the proximal and distal halves of the stent relative to the closest reference segment [5]. It has been observed that OCT provides more favorable predictors of stent underexpansion than angiography, including calcium burden, area and volume [6]. Stent underexpansion was found to be potentially related with clinical outcomes in patients treated with PCI in terms of greater frequency of target lesion revascularization (TLR) [7,8]. Additionally, newer OCT-derived methods for volumetric assessment of stent expansion were found to be better within the aspect of lower TLR occurrence during the follow-up period [7]. Some authors have attempted to indicate thresholds of maximal calcium thickness predicting acceptable stent expansion, which was defined as >80% in that study and found that a calcium thickness <880 m was a useful predictor for acceptable stent expansion in moderate calcified lesions treated with PCI and without debulking devices [9]. Therefore, several calcium scoring systems were invented in the past to extract patients having lesions with a high possibility of stent underexpansion following PCI [10]. Among factors related to the poorer prognosis of stent underexpansion they found maximal calcium angle >180º, maximum calcium thickness >0.5 mm and length >5 mm. This exemplary calcium scoring system was invented to extract patients demanding calcium plaque modification with mechanical or ultrasound devices [10]. 

In the current study we aimed to compare the processing results, procedural characteristics and clinical outcomes according to the extent of stent expansion, as well as risk factors of stent underexpansion in patients treated with PCI using OCT.

## 2. Methods

### 2.1. Patients

The current analysis included 100 consecutive patients presenting with stable angina, who were treated with PCI assisted by OCT between June 2021 and April 2022, which was performed at the Department of Cardiology and Cardiovascular Interventions University Hospital in Kraków, Poland. The decision about the use of OCT was left to the first operator, who performed the PCI. Then, patients were divided into 2 groups according to the extent of stent expansion (29 patients for <90% stent expansion and 71 patients for ≥90% stent expansion), and we assumed the limit value of 90% according to the currently accepted limit for optimal stent expansion in the case of OCT-guided PCI [11].

### 2.2. Optical Coherence Tomography—Image Aquisition and Processing

All PCIs were guided with OCT equipped with the newest OPTIS™ Integrated Next Imaging System unit with Ultreon™ 1.0 Software (Abbott Vascular, Santa Clara, CA, USA). The only OCT catheter applied during the use of that equipment was the Dragonfly™ OPTIS™ Imaging Catheter (Abbott Vascular, Santa Clara, CA, USA). The procedure of acquisition of intravascular images using the OCT probe was carried out in accordance with recent international recommendations and was repeated at least twice, i.e., pre—and in selected cases, i.e., chronic total occlusions or tight stenosis, during the final stage of the procedure. In the majority of cases, however, 3 runs took place: pre—before predilatation, during—after predilatation or before postdilatation, and post—after stent implantation and postdilatation. The Dragonfly™ OPTIS™ Imaging Catheter was connected to a dedicated system. We used OCT-angiography co-registration. The pullback length was adjusted to the length of the visualized artery: 54 mm or 75 mm. In selected cases, it had to be repeated twice due to lesions longer than 75 mm. Pullback was performed automatically. A contrast medium (VisipaqueTM, Iodixanol 320 mg/mL, GE Healthcare) was used to clear the lumen of blood, making it possible for the OCT to perform imaging. Manual injections were adopted in all cases with 10 or 20 cc syringes, according to the type of artery (diameter and length of lesion). Each contrast flush was preceded by 200 µg of nitro-glycerin intracoronary injection. Flush clearance allowed for distinction between the lumen and structure of the vessel. Final assessment was carried out using Ultreon 1.0TM Software according to the Morphology, Length, Diameter, Medial Dissection, Apposition and Expansion (MLD-MAX) algorithm recommended by Abbott Vascular. Several records were deleted due to lack of pre- or post-stenting examination, or none, which may have been the case, for example, in probe damage at baseline (while preparing the catheter or during the examination. Furthermore, in the instance of using the OCT probe after PCI for various reasons, e.g., complications, difficulties with final procedure result evaluation (coronary angiography alone or with intravascular ultrasound) or to visualize the endovascular procedure, this idea was born during the procedure. Cases were also removed from analysis in situations when any of the records prevented the assessment of the artery before or after stenting due to poor quality, e.g., insufficient contrast washout and thus, resulting in the inability to calculate vessel lumen or external elastic lamina (EEL) before PCI, as well as stent expansion following the implantation. For this reason, a few patients were removed from the analysis, which comprising less than 10% of all patients participating in the pre-analysis of the current study. In the case of satisfactory recordings, the analysis of OCT images were performed by at least 1 or 2 highly experienced operators during the procedure, and another blinded analysis by an experienced and blinded operator in the evaluation of OCT images, who finally analyzed all cases before entering them into the database. Adding diagnostic ability of plaque characteristic and easier planning of stenting strategy using artificial intelligence are among the main innovations of the Ultreon 1.0TM Software. In the current analysis, prior to PCI, we assessed plaque type, maximal calcium angle, maximal calcium thickness and length, minimal lumen diameter (MLD), distal reference EEL diameter and distal lumen diameter (in the case of lacking EEL). After PCI, we assessed medial dissection, malapposition, minimum stent expansion and MLD. Then, we additionally calculated the distal reference EEL diameter to minimum stent diameter, distal lumen diameter (in the case of no EEL) to minimum stent diameter and MLD after PCI to minimum stent diameter ratios. The comparative use of Ultreon™1.0 Software with the previous generation software has been assessed by our team in our recently published study [12]. The study was conducted in accordance with the Declaration of Helsinki, there were no approval of the Regional Bioethics Committee, due to the retrospective nature of the study.

### 2.3. Definitions

The main endpoint relating to OCT use comprised predictors of minimal mean stent expansion, assessed automatically by the Ultreon™1.0 Software. In the majority of cases, the minimal stent expansion diameter was assessed by Ultreon™ 1.0 Software, however, in each instance, the result was manually checked and adjusted frame-by-frame, if necessary. Incorrect measurements, e.g., in cases of insufficient contrast clearance or deviation from the oval cross-section of the artery, which can cause problems with correct tracing of the artery lumen by the software, were corrected manually.

Other study endpoints of the current analysis included the relationship between the degree of stent expansion and clinical outcomes, which included major adverse cardiovascular events (MACE), considered as the following: cardiac death, myocardial infarction (MI), revascularization: either surgical or percutaneous (Re-PCI; TLR; TVR, target vessel revascularization) and/or cerebrovascular events, e.g., stroke or transient ischemic events. Device-oriented composite endpoints, including cardiac death, target vessel related myocardial infarction (TV-MI), as well as TLR, were also evaluated.

Kidney failure was defined for the purposes of the current publications as glomerular filtration rate lower than 60 mL/kg/min. Hypercholesterolaemia was defined according to the current European guidelines [13].

### 2.4. Statistical Analysis

Categorical variables are given as numbers and percentages. Continuous variables are presented as means ± standard deviations, or as medians and interquartile ranges in the case of non-normal data distribution. Normality of distribution was analyzed using the Shapiro–Wilk test. Equality of variance was subjected to evaluation by implementing Levene’s test. The differences noted between groups were subject to comparison via the Student’s or Welch’s *t*-tests. This was dependent on the equality of variance regarding normally-distributed variables. The Mann–Whitney U test was used for continuous variables lacking normal distribution. Categorical variables were also compared via Pearson’s chi-squared or Fisher’s exact test. This was done when 20% of cells had a count below 5. Ordinal variables were further compared via the Cochran–Armitage trend test. All baseline, demographic, and procedural characteristics were assumed as possible predictors of stent expansion in simple linear models, as well as simple logistic regression models with stent expansion <90% as dependent variable. Due to the number of patients in each group, we were not able to construct multivariable regression analysis models. The study has 80% power to estimate the proportion of stent expansion <90 with 15% precision assuming worst case scenario (50%) at 5% two-sided significance level. Additionally, Kaplan–Meier survival estimates were constructed for comparison of mortality and MACE occurrence during the follow-up period, and a log-rang test was conducted to calculate significance of differences. All of the performed statistical analyses were carried out using JMP^®^, Version 16.1.0 (SAS Institute INC., Cary, NC, USA, 2021).

## 3. Results

### 3.1. General Characteristics and Concomitant Disease at Baseline (before Index Procedure)

Patients from the stent underexpansion group underwent percutaneous interventions on the peripheral arteries significantly more often (*p* = 0.02). There were no other significant differences between the groups (Table 1).

### 3.2. Biochemical Characteristics at Baseline (before Index Procedure)

There were no significant differences in selected blood parameters between both groups of patients, except for the pre-PCI serum creatinine concentration (*p* = 0.02) (Table 2).

### 3.3. Procedural Characteristics (Index Procedure)

There were no significant differences in selected procedural indices between both groups of patients (Table 3).

### 3.4. OCT Parameters (Index Procedure)

There were no significant differences in selected OCT measurements between both groups, except for mean minimum stent expansion, which was assumed to be significantly different between the 2 groups (Table 4).

### 3.5. Pharmacotherapy (after Index Procedure)

There were no significant differences in pharmacological treatment between both groups of Table 5.

### 3.6. Clinical Outcomes

There were no significant differences in clinical outcomes or mean length of the follow-up between both groups (Table 6). Kaplan–Meier survival estimates are presented in Figure 1A,B.

### 3.7. Predictors of Stent Underexpansion—Simple Linear Univariable Analysis

Significant predictors of stent underexpansion assessed by simple linear univariable analysis included: hypercholesterolemia, obstructive bronchial diseases and treatment with inhalators, family history of cardiovascular disease, PCI of other than the left main coronary artery (LMCA), stent and drug-eluting stent implantation, PCI without drug-eluting balloon, paclitaxel antimitotic agent, greater maximal stent diameter, and lower Euroscore II (Table 7).

### 3.8. Predictors of Stent Underexpansion—Logistic Regression Univariable Analysis

Univariable logistic regression analysis confirmed a correlation between stent underexpansion and greater creatinine serum concentration before and after PCI (Table 8, Figure 2).

## 4. Discussion

Among the main findings of the current study, was that there were no significant differences in general characteristics between both groups at baseline, except for the greater occurrence of percutaneous peripheral interventions and greater mean serum pre-PCI creatinine concentration in the stent underexpansion group. Secondly, the current analysis revealed that significant predictors of stent underexpansion assessed by simple linear univariable analysis included: hypercholesterolemia, obstructive bronchial diseases and treatment with inhalators, family history of cardiovascular disease, PCI of other than the LMCA, stent and drug-eluting stent implantation, PCI without drug-eluting balloon, paclitaxel antimitotic agent, greater maximal stent diameter and lower Euroscore II. Thirdly, univariable logistic regression analysis confirmed a correlation between stent underexpansion and greater creatinine serum concentration before and after PCI.

UltreonTM 1.0 Software uses artificial intelligence algorithms to analyze the OCT image. The program automatically indicates locations of severe calcifications that require modification before stent implantation, but also locations where the artery does not have significant atherosclerotic changes, i.e., locations free from atherosclerotic plaque. The software not only facilitates the assessment of the vessel morphology, but also analyzes the internal dimensions of the vessel, detects the EEL – calculating media to media size. Knowing these dimensions and the morphology of the artery, you can immediately make a decision whether or not to modify the atherosclerotic plaque and establish the proximal and distal reference dimensions of the stented segment. What is, in our experience, extremely important, is that the OCT examination helps in the optimization of the stent, because it indicates the locations of stent malapposition and suggests where the stent is insufficiently expanded. UltreonTM 1.0 Software is therefore a user friendly tool for OCT image interpretation and decision making. Importantly, when used skillfully, OCT does not increase the consumption of contrast and may contribute to the reduction of radiation by eliminating unnecessary projections during the procedure.

Considering the differences between both analyzed groups, it could be expected that in the group with a worse final result of stent expansion, there will be significantly more factors potentially related to the presence of, e.g., coronary calcifications, renal failure, diabetes, previous percutaneous and surgical revascularization procedures, age, male sex and a number of other issues, e.g., greater increase in calcification in OCT. Such differences are present in the analyzed study, but the decisive factor for the lack of significance is undoubtedly the small group of patients. It may be expected that in the case of a large group of patients, some of these factors would reach statistical significance. The only factor that reached statistical significance was the more frequent number of percutaneous peripheral interventions in the stent underexpansion group assessed by OCT, which may be indirectly related to more advanced atherosclerosis of the coronary arteries and their greater stiffness related to, for instance, a higher incidence and severity of calcification. 

In previously published studies, it has been demonstrated that the maximum arc of target lesion calcification is a predictor of stent expansion and, therefore, it is the minimal stent area possible to attain [14,15]. This was also found to be a strong predictor regarding outcomes following the implantation of drug-eluting stents [16,17]. Another factor undergoing more advanced analysis in recent years has been the presence of calcium fracture following lesion preparation [18]. In the current study more than 12% of patients were treated with intravascular lithotripsy and another 35% with rotablation, which could blur the significance of relationship between arterial calcifications and extent of stent expansion. 

Moving on to the discussion of factors related to the lower likelihood of satisfactory stent expansion, which turned out to be important in the analysis conducted for the purpose of this publication, several factors typical for increased risk of atherosclerosis development and its faster progression, and an indirect relationship with greater calcification of lesions, were among them: hypercholesterolemia, chronic obstructive pulmonary disease, treatment with inhalators and a family history of cardiovascular disease. Chronic obstructive pulmonary disease and treatment with inhalators is undoubtedly associated with smoking, which is definitely a strong risk factor for atherosclerosis.

PCI within LMCA was found to be a factor favoring better stent expansion, which seems to be justified as the diameter of the LMCA is usually larger compared to other vessels. This is further associated with the implantation of larger-sized stents. Additionally, in the case of single-stenting of the LMCA and anterior descending and/or circumferential or intermediate branches, there is a connection with the change in vessel diameter in the longitudinal section. This is rarely determined as stent underexpansion by the software, not even by the most innovative versions. In the case of such an assessment, underexpansion is easy to eliminate with additional post-dilatation.

Observations regarding the relationship between stent expansion and the stented artery (LMCA) were closely related to another factor, namely the maximum diameter of the stent. Larger diameter stents were less likely to be under-expanded after PCI in OCT assessment.

The next 3 predictors of stent expansion stated as statistically significant included PCI with stent implantation, the use of PCI with DEB, and implementation of the antimitotic drug paclitaxel, which was found only in antimitotic drug-coated balloons, specifically, in 1 patient. Therefore, the results should be interpreted with caution and require confirmation on much larger groups of patients. In the case of PCI with DEB, it is also important to prepare the lesion (in-stent restenosis) by proper pre-dilation, which determinates final OCT assessment and percentage of stent expansion. An unexpected finding of the presented study seems to be the fact that lower Euroscore II value turned out to be a predictor of worse stent expansion. Nevertheless, in univariable regression analysis, it turned out that only the level of creatinine before and after PCI remains a predictor of poorer stent expansion, which, considering the fact that there were no dialysis patients in the analyzed group, seems to be a strong relationship. In the case of dialysis patients, such a relationship would seem much more obvious due to the very high risk of calcification and stiffening of the arteries, increasing their fragility, worsening vascular accessibility, and maintaining correct positioning of the guiding catheter. In most studies on stent expansion, predictors are for intravascular ultrasound PCI, while OCT studies are less frequently published because its use is often restricted to small groups, while statistical analyzes conducted to identify viral stent under-expansion predictors require large patient cohorts [19]. It should be noted that in the analyzed study, most stents were properly expanded, which in many cases required the use of 2 plaque debulking devices, such as rotablation and orbital atherectomy or rotablation and intravascular lithotripsy for better lesion preparation. If the group of under-expanded stents had been larger and had included more patients with large average stent under-expansion, the differences would have been greater between both groups and predictors of stent under-expansion would be easier to detect. Nonetheless, it is worth emphasizing that in the current era, the percentage of satisfactory expanded stents in patients treated with PCI assisted by OCT is high, despite the initial experience with the latest software at our catheterization laboratory.

## 5. Conclusions

Based on the presented analysis, the degree of stent expansion is not related to procedural factors, OCT imaging indices or clinical outcomes. Logistic regression analysis confirmed such a relationship only for creatinine level assessed before and after PCI. The fact that the Ultreon 1.0^TM^ Software does not improve stent expansion depends mainly on individual operators, but it is mainly a tool that facilitates the procedure, which allows shortening the treatment time among operators who use OCT less frequently and occasionally compared to very experienced operators who use it on a daily basis, while maintaining a high level of treatment.

## 6. Limitations

Undoubtedly, the presented results are preliminary. The study group is limited to a small size, and strong conclusions cannot be drawn. The presence of a learning curve among individual operators is of great importance, and the initial results of the first treatments may differ significantly from subsequent, systematized treatments. Due to the small group of patients, participants were not matched using propensity score matching, which would take a number of factors, including patient age, concomitant diseases, vessel diameter, degree of calcification, or the urgency of the procedure, and a number of others into account. In the presented study, preliminary results are shown, and research will be continued as the group of patients grows. The Ultreon 1.0^TM^ Software enables the performance of more complicated procedures in more complex patients who, in many cases, would be qualified for conservative treatment. Artificial intelligence, helping to improve percutaneous interventions, contributes to shortening procedure duration and improving its results. Another issue is that the characteristics of patients who undergo OCT intravascular imaging are constantly evolving, and it is extremely difficult to collect a large group of patients in a short time to obtain reliable results, at least at the current stage of OCT prevalence, despite the fact that it is exponentially increasing in Poland, if only due to the fact that the costs of its use have been reimbursed by the National Health Fund since January 2022.

## Figures and Tables

**Figure 1 medicina-58-01227-f001:**
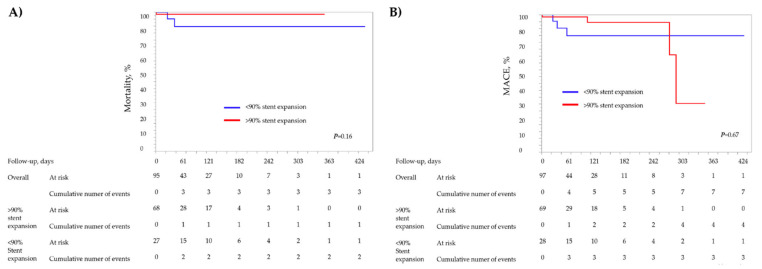
Kaplan–Maier survival estimates of mortality and MACE occurrence during follow-up period according to the stent expansion. (**A**) Kaplan–Meier survival estimates for mortality. (**B**) Kaplan–Meier survival estimates for MACE.

**Figure 2 medicina-58-01227-f002:**
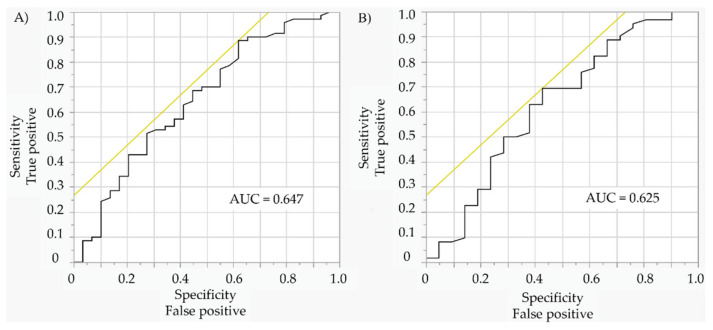
(**A**) Receiver operating characteristic curve of the relationship between pre-procedural creatinine serum concentration and stent expansion (≥90% vs. <90%). (**B**) Receiver operating characteristic curve of the relationship between post-procedural creatinine serum concentration and stent expansion (≥90% vs. <90%).

**Table 1 medicina-58-01227-t001:** General characteristics and concomitant diseases.

	Total N = 100	Stent Expansion <90% N = 29	Stent Expansion ≥90% N = 71	*p*-Value
Age, years	67.9 (62.3; 75.5)	67.4 (62.3; 74.6)	68.2 (62.4; 75.6)	0.66
Gender, male	80 (80)	26 (89.7)	54 (76.1)	0.1
Euroscore II, %	2.3 (1.2; 3.5)	2.4 (1.4; 4.7)	2.2 (1.2; 3.3)	0.22
STS score	1.6 (0.8; 3)	2.1 (0.9; 3.5)	1.4 (0.7; 2.8)	0.24
Syntax I	15.5 (8; 24)	14 (8; 24.8)	16 (9; 24)	0.92
Syntax II PCI 4-year mortality, %	15.2 (6.8; 29.9)	23.5 (8.7; 42.6)	10.8 (5.9; 23.8)	0.08
Syntax II CABG 4-year mortality, %	11 (4.8; 17.4)	12.9 (5; 22.6)	10.4 (4.8; 14.6)	0.29
Diabetes mellitus	37 (37)	10 (34.5)	27 (38)	0.74
Hypercholesterolemia	56 (56)	17 (58.6)	39 (54.9)	0.68
Arterial hypertension	83 (83)	25 (86.2)	58 (81.7)	0.58
Kidney failure	7 (7)	2 (6.9)	5 (7)	0.98
Dialysis therapy	1 (1)	1 (3.5)	0 (0)	0.11
Prior PCI	58 (58)	19 (65.5)	39 (65.9)	0.33
Prior CABG	4 (4)	1 (3.5)	3 (4.2)	0.86
COPD/Bronchial asthma	16 (16)	7 (24.1)	9 (12.7)	0.17
Smoking	23 (23)	8 (27.6)	15 (21.1)	0.61
Family history of CVD	11 (11)	5 (17.2)	6 (8.5)	0.22
Heart failure	59 (59)	18 (62.1)	41 (57.8)	0.69
LVEF, %	40 (20.5; 53.8)	38 (20; 55.5)	40 (23; 50)	0.69
Peripheral arterial disease	14 (14)	7 (24.1)	7 (9.9)	0.07
Prior PTA	2 (2)	2 (6.9)	0 (0)	0.02
Prior stroke	7 (7)	3 (10.3)	4 (5.6)	0.42

Data are expressed as mean ± standard deviation and median ÷ interquartile range where necessary (non-normal distribution) or numbers (percentages). CABG, coronary artery bypass grafting; COPD, chronic obstructive pulmonary disease; CVD, cardiovascular disease; LVEF, left ventricle ejection fraction; PCI, percutaneous coronary intervention; PTA, percutaneous transluminal angioplasty; STS, Society of Thoracic Surgeons.

**Table 2 medicina-58-01227-t002:** Biochemical indices.

	Total N = 100	Stent Expansion <90% N = 29	Stent Expansion ≥90% N = 71	*p*-Value
Creatinine before PCI, µmol/L	90.5 (73.9; 106)	99.1 (83.1; 123)	87.7 (72.1; 102)	0.02
Creatinine after PCI, µmol/L	90.2 (78; 117)	105 (82.9; 137)	89.1 (77.7; 110.3)	0.09
GFR, mL/min.	74 (60; 83)	64 (52.5; 80)	76 (64; 84)	0.11
PLT, × 10^3^/µL	235.5 (185; 310)	246 (199; 346.5)	227 (185; 295)	0.22
Hemoglobin before PCI, mg/dL	13.7 ± 1.8	13.3 ± 1.7	13.9 ± 1.8	0.26
Hemoglobin after PCI, mg/dL	12.6 ± 2.2	12.6 ± 2.4	12.6 ± 2.1	0.89
RBC, × 10^6^/µL	4.5 ± 0.6	4.4 ± 0.5	4.5 ± 0.6	0.67
CRP, mg/dL	10 (2.6; 19)	8.9 (2.6; 17.8)	10.3 (2.6; 20.4)	0.94
Total cholesterol, mmol/L	3.8 (3.3; 4.8)	3.7 (3.2; 4.7)	3.9 (3.3.; 4.8)	0.49
LDL cholesterol, mmol/L	2 (1.5; 2.6)	1.8 (1.5; 2.6)	2 (1.5; 2.7)	0.45
TGL, mmol/L	1.4 (1; 1.8)	1.3 (0.9; 1.7)	1.4 (1.1.; 1.9)	0.55
HDL cholesterol, mmol/L	1.1 (1; 1.3)	1.2 (0.9; 1.4)	1.1 (1; 1.3)	0.63
Maximal CK-Mb, IU/L	22.5 (14.8; 46.3)	20.5 (16; 25.3)	22.5 (14; 48.8)	0.6
Maximal troponin T, pg/mL	274 (65.1; 1307)	246 (92.5; 829)	274 (63.7; 1646)	0.47
NT-pro BNP, pg/mL	786 (773; 3312)	2636 (773; 5287)	1673 (740; 3008)	0.27
Glucose, mmol/L	6.2 (5.4; 8.3)	8.3 (6; 9.4)	6.1 (5.3; 7.5)	0.14

Data are expressed as mean ± standard deviation and median ÷ interquartile range were necessary (non-normal distribution) or numbers (percentages). BNP, B-type natriuretic peptide; CK-Mb, creatine kinase myocardial bound; CRP, C-reactive protein; GFR, glomerular filtration rate; HDL, high-density lipoproteins; LDL, low-density lipoproteins; NT, N-termina; PCI, percutaneous coronary intervention; PLT, platelets count; RBC, red blood cells; TGL, triglycerides.

**Table 3 medicina-58-01227-t003:** Procedural charcteristics.

	Total N = 100	Stent Expansion <90% N = 29	Stent Expansion ≥90% N = 71	*p*-Value
LMCA	34 (34)	6 (20.7)	28 (39.4)	0.07
LAD	78 (78)	22 (75.9)	56 (78.9)	0.74
Diagonal branch	16 (16)	7 (24.1)	9 (12.7)	0.17
Circumflex branch	33 (33)	8 (27.6)	25 (35.2)	0.46
Marginal branch	6 (6)	2 (6.9)	4 (5.6)	0.81
Right coronary artery	28 (28)	7 (24.1)	21 (29.6)	0.58
Chronic total occlusion	10 (10)	2 (6.9)	8 (11.3)	0.49
PCI + stent	97 (97)	29 (100)	68 (95.8)	0.15
Drug-eluting balloon	3 (3)	0 (0)	3 (4.2)	0.15
Drug-eluting stent	97 (97)	29 (100)	68 (95.8)	0.15
Bioresorbable scaffold	0 (0)	0 (0)	0 (0)	-
Type of antimitotic agent:				
Ewerolimus	87 (87)	25 (86.2)	62 (87.3)	0.88
Sirolimus	13 (13)	4 (13.8)	9 (12.7)	0.88
Zotarolimus	3 (3)	1 (3.5)	2 (2.8)	0.87
Number of stents	2 (1; 3)	2 (1; 2.5)	2 (1; 3)	0.87
Number of stents				0.31
0	3 (3)	0 (0)	3 (4.2)	
1	26 (26)	9 (31)	17 (23.9)	
2	43 (43)	13 (44.8)	30 (42.3)	
3	14 (14)	2 (6.9)	12 (16.9)	
4	11 (11)	3 (10.3)	8 (11.3)	
5	3 (3)	2 (6.9)	1 (1.4)	
Overall stent length, mm	56 (33; 76)	50 (36; 63.5)	56 (33; 76)	0.39
Maximal stent diameter, mm	3.5 (3; 3.5)	3.5 (3; 3.5)	3.5 (3; 3.5)	1
Minimal stent diameter, mm	3 (2.5; 3)	2.8 (2.5; 3)	3 (2.5; 3.4)	0.16
Maximal balloon diameter, mm	4 (3.5; 5)	3.6 (3.5; 4.5)	4 (3.5; 5)	0.26
Maximal balloon pressure, atm.	20 (18; 24)	22 (18; 25)	20 (18; 24)	0.76
Rotablation	27 (27)	10 (34.5)	17 (23.9)	0.29
IVL	12 (12)	2 (6.9)	10 (14.1)	0.29
Impella pump	35 (35)	13 (44.8)	22 (31)	0.19
IABP	3 (3)	0 (0)	3 (4.2)	0.15

Data are expressed as mean ± standard deviation and median ÷ interquartile range where necessary (non-normal distribution) or numbers (percentages). IABP, intra-aortic balloon pump; IVL, intravascular lithotripsy; LAD, left anterior descending artery; LMCA, left-main coronary artery; PCI, percutaneous coronary intervention.

**Table 4 medicina-58-01227-t004:** Optical coherence tomography parameters.

	Total N = 100	Stent Expansion <90% N = 29	Stent Expansion ≥90% N = 71	*p*-Value
Before PCI				
Type of plaque				0.56
- lipidic	10 (10)	2 (6.9)	8 (11.3)	
- fibrotic	13 (13)	3 (10.3)	10 (14.1)	
- mild/moderate calcium	26 (26)	6 (20.7)	20 (28.2)	
- severe calcium	51 (51)	18 (62.1)	33 (46.5)	
Maximum calcium angle, °	189 (93; 299)	230 (122; 299)	181 (0; 300.3)	0.41
Maximal calcium thickness, mm	1.1 (0.6; 1.4)	1.1 (0.7 ± 1.4)	1.1 (0; 1.4)	0.71
Total calcium length, mm	5 (2; 20)	8 (3; 20)	4 (0; 20)	0.28
Minimal lumen diameter, mm	1.4 (1.2; 1.7)	1.4 (1.1; 1.6)	1.4 (1.3; 1.8)	0.43
Distal EEL reference diameter, mm	3.3 ± 0.7	3.3 ± 0.7	3.3 ± 0.7	1
Distal EEL reference to minimum stent diameter ratio	1.1 (1; 1.2)	1.1 (1; 1.2)	1.1 (1; 1.2)	0.71
Distal lumen diameter, mm	2.3 (1.9; 2.9)	2.4 (1.9; 2.8)	2.3 (1.9; 2.9)	0.91
Distal lumen to minimum stent diameter ratio	0.8 (0.7; 0.9)	0.8 (0.7; 0.9)	0.8 (0.7; 0.9)	0.41
After PCI				
Medial dissection	7 (7)	1 (3.5)	6 (8.5)	0.34
Malapposition	3 (3)	2 (6.9)	1 (1.4)	0.17
Minimum stent expansion, %	95 (87.3; 104)	82 (75.5; 85.5)	101 (94; 112)	<0.001
Minimal lumen diameter, mm	2.6 (2.3; 3)	2.5 (2.1; 2.9)	2.7 (2.3; 3.1)	0.19
Minimal lumen to minimum stent diameter ratio	0.9 (0.8; 1)	0.9 (0.8; 1)	0.9 (0.8; 1)	0.34

Data are expressed as mean ± standard deviation and median ÷ interquartile range where necessary (non-normal distribution) or numbers (percentages) EEL, external elastic lamina; PCI, percutaneous coronary intervention.

**Table 5 medicina-58-01227-t005:** Pharmacotherapy.

	Total N = 100	Stent Expansion <90% N = 29	Stent Expansion ≥90% N = 71	*p*-Value
Acetyl-salicylic acid	96 (96)	28 (96.6)	68 (95.8)	0.86
P2Y12 inhibitor				
- clopidogrel	49 (49)	14 (48.3)	35 (49.3)	0.93
- prasugrel	13 (13)	2 (6.9)	11 (15.5)	0.22
- ticagrelor	37 (37)	13 (44.8)	24 (33.8)	0.30
Antithrombotic treatment(heparins, old and new oral anticoagulants)	17 (17)	4 (13.8)	13 (18.3)	0.58
Lipid-lowering treatment				
- statin	95 (95)	28 (96.6)	67 (94.4)	0.64
- fibrate	2 (2)	1 (3.5)	1 (1.4)	0.53
- ezetimibe	12 (12)	3 (10.3)	9 (12.7)	0.74
Insulin therapy	18 (18)	6 (20.7)	12 (16.9)	0.66
Oral anti-diabetic therapy	30 (30)	6 (20.7)	24 (33.8)	0.18
Angiotensin converting enzyme inhibitors	80 (80)	24 (82.8)	56 (78.9)	0.66
Calcium channel blockers	21 (21)	9 (31)	12 (16.9)	0.13
Beta blocker	86 (86)	25 (86.2)	61 (85.9)	0.97
Inhalators (bronchodilators/GCSs)	11 (11)	6 (20.7)	5 (7)	0.06

Data are expressed as mean ± standard deviation and median ÷ interquartile range where necessary (non-normal distribution) or numbers (percentages). GCSs, glucocorticosteroids.

**Table 6 medicina-58-01227-t006:** Clinical outcomes.

	Total N = 100	Stent Expansion <90% N = 29	Stent Expansion ≥90% N = 71	*p*-Value
Completed follow-up	99 (99)	29 (100)	70 (98.6)	0.41
TLR	1 (1)	0 (0)	1 (1.5)	0.41
TVR	1 (1)	0 (0)	1 (1.5)	0.41
Myocardial infarction	1 (1)	1 (3.6)	0 (0)	0.11
Stroke	2 (2)	0 (0)	2 (2.9)	0.24
Re-PCI	3 (3)	2 (7.1)	1 (1.5)	0.17
CABG	0 (0)	0 (0)	0 (0)	-
DOCE	5 (5.1)	2 (7.1)	3 (4.3)	0.57
Cardiac death	4 (4)	2 (7.1)	2 (2.9)	0.36
TV-MI	0 (0)	0 (0)	0 (0)	-
MACE	10 (10)	4 (13.8)	6 (8.5)	0.43
Death overall	6 (6)	3 (10.3)	3 (4.2)	0.26
Mean follow-up duration, days	42 (22; 136.3)	65 (23; 177)	41 (20; 126)	0.24

Data are expressed as mean ± standard deviation and median ÷ interquartile range where necessary (non-normal distribution) or numbers (percentages) CABG, coronary artery bypass grafting; DOCE, device-oriented composite endpoint; MACE, major adverse cardiac events; Re-PCI, repeat percutaneous coronary intervention; TLR, target lesion revascularization; TVR, target vessel revascularization.

**Table 7 medicina-58-01227-t007:** Predictors of stent underexpansion—simple linear univariable analysis.

Variable	Estimate	95% Confidence Interval	*p*-Value
Hypercholesterolemia, 1 vs. 0	−28.16	−40.34–(−15.98)	<0.01
COPD/Bronchial asthma, 0 vs. 1	6.19	1.05–11.34	0.02
Familiar history of CVD, 0 vs. 1	6.24	0.16–12.31	0.04
PCI within LMCA, 0 vs. 1	−4.45	−8.45–(−0.45)	0.03
Stent implantation, 0 vs. 1	11.7	0.56–22.84	0.04
PCI with DEB, 0 vs. 1	−11.7	−22.84–(−0.56)	0.04
Stent type, DES, 0 vs. 1	11.7	0.56–22.84	0.04
Paclitaxel antimitotic agent vs. other	−40.25	−58.02–(−22.49)	<0.001
Inhalators, 0 vs. 1	6.64	0.58–12.7	0.03
Euroscore II, %	1.41	0.04–2.77	0.04
Maximal stent diameter, mm	−1.24	−2.41–(−0.06)	0.04

CABG, coronary artery bypass grafting; COPD, chronic obstructive pulmonary disease; CVD, cardiovascular disease; GKS, glucocorticosteroids; LDL, low-density lipoproteins; NT-proBNP, N-terminal-pro-B-type natriuretic peptide; TGL, triglycerides; STS, Society of Thoracic Surgeons.

**Table 8 medicina-58-01227-t008:** Predictors of stent expansion—logistic regression analysis.

Variable	Odds Ratio	95% Confidence Interval	*p*-Value
Creatinine before PCI	0.97	0.95–0.99	0.01
Creatinine after PCI	0.98	0.96–0.99	0.02

PTA, percutaneous transluminal angioplasty.

## Data Availability

The analyzed source data may be made available on a special, justified request, which should be sent to the following address: jaanraf@interia.pl.

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
