# Peer review of "Experience with Optical Coherence Tomography Enhanced by a Novel Software (Ultreon™ 1.0 Software)—The First One Hundred Cases"

_medicina, 2022, doi:10.3390/medicina58091227_

Round 1
Reviewer 1 Report
I reviewed with interest the article submitted. Although interesting, there are some concerns related to the interpretation and presentation of the data that require a large revision of the manuscript.
1. Multivariable logistic regression analysis demonstrated a correlation between poorer stent expansion and creatinine serum concentration. Could authors provide creatinine levels (mean and SD) before and after PCI?
2. Could authors provide Kaplan-Meier curves to assess event-free survival during FUP with possible difference between two groups?
3.Polyvascular atherosclerotic disease (PAD) has emerged as an aggressive form of atherosclerosis, related to worse outcome in patients with coronary artery disease (10.1016/j.numecd.2021.09.032). As authors showed, peripheral artery disease and previous peripheral revascularization are related to poorer stent expansion assessed by OCT, probably due to greater calcification and/or complexity of plaque even in coronary district. Could OCT images confirm this hypothesis?
4. English style and language have to be extensively improved. Several mistakes are present; some concepts are not completely clear.
5. The “2.2. Optical coherence tomography – image attainment and processing” section is not adequate at all. Considering the relevant role of this section about the methodological approach of the research, more clearness is needed.
6. There are different problems in text editing. Section “2.3. Definitions” and description of Figure 1 show a different format compared with main text.
7. Check the words spelling in Table 1 (hipercholesterolemia, dyalisotherapy, etc).
8. It is more appropriate to use “poorer stent expansion group” rather than “patients from group of patients with poorer stent expansion”.
9. In Tables, the insertion of median ÷ interquartile range is not necessary.
10. Check the description in Figure 1. What does “(< 90 ≥ %)” stand for?
Author Response
Reviewer 1
The manuscript “Experience with Optical Coherence Tomography Enhanced by Novel Software (Ultreon 1.0TM) – the First One Hundred Cases”, although on a low number of cases, is looking for predictors of stent underexpansion using OCT as a recognized tool.
The authors compare the “processing results, procedural indices, clinical outcomes according to the extent of stent expansion and assess risk factors of poorer stent expansion in patients treated with PCI using OCT” The technical work is high (they assessed each frame of 100 patients, with 2-3 pullbacks per patients) to quantify the atherosclerotic plaques and to define the stents expansion).
The authors describe the method very accurate, the statistical analysis is well done.
The fact that the procedural characteristics are not predictors means the operators are extremely skilled and no differences are among the two groups.
The results are interesting as predictors for poor expansion are finallyrefering only to the creatinine level.
I consider this research and manuscript acceptable for publication, with the condition to answer to several comments:
- Please, state clearly at the beginning of the discussion section, the originality and the importance of the study.
A relevant paragraph was added at the beginning of the discussion.
- Please, mention the AIC and BIC of the models analysed.
This has been added in the Table 5 and 8.
- Please calculate and mention the statistical power of the study (1-β).
This was included into the Statystical analysis section.
- Please, mention at the discussion section, why time of balloon inflation was not taken into consideration.
Yours sincerely for the question, but I do not really understand what kind of duration is involved, when the beginning and when the end is, and what this would bring to the present publication, i.e. what aspect of the discussion would be included in this information.
- Please, explain what type of balloons were used. High pressure balloon (Schwager) was taken into consideration to prepare the culprit lesions (Secco, Di Mario et al., EuroIntervention, 2016)?
The balloon hardness was adjusted to the needs, i.e. the degree of calcification of the vessel and its susceptibility to balloon expansion, depending on the operator's decision, semi-complient, non-complient or non-complient high-pressure ballons were used. This sentence was added to the discussion. I do not have precise data on which procedure, which balloons and in what sequence were used, but this is not the main point of this publication.
- Please, consider adding the measurement units for creatinine (table).
This was added (Three extra Table were moved from suplementarny materials to the main manuscript section)
- Please, consider only mean±SD or median (IQR) for each variable, not both. When the characteristic in one group is abnormally distributed use median as the central tendency.
This was corrected.
- Please, describe the definition of hypercholesterolemia used for the study.
An appropriate reference was added.
- Please, define kidney failure (table 1). What is the degree of the CKD?
This was added into the Methods section.
- Please, consider to present the multivariable analysis for all the variables identified as predictors at the univariate logistic regression analysis, not only for creatinine.
This was modified, due to the fact that there were no multivariable analysis.
- What is the sensitivity and specificity for creatinine, and what is the cut-off of the predictor, if calculated?
This was added into the results section point: 3.8.
Reviewer 2
I reviewed with interest the article submitted. Although interesting, there are some concerns related to the interpretation and presentation of the data that require a large revision of the manuscript.
- Multivariable logistic regression analysis demonstrated a correlation between poorer stent expansion and creatinine serum concentration. Could authors provide creatinine levels (mean and SD) before and after PCI?
This was added and modified. There were no multivariable regression analysis.
- Could authors provide Kaplan-Meier curves to assess event-free survival during FUP with possible difference between two groups?
This has been added.
3.Polyvascular atherosclerotic disease (PAD) has emerged as an aggressive form of atherosclerosis, related to worse outcome in patients with coronary artery disease (10.1016/j.numecd.2021.09.032). As authors showed, peripheral artery disease and previous peripheral revascularization are related to poorer stent expansion assessed by OCT, probably due to greater calcification and/or complexity of plaque even in coronary district. Could OCT images confirm this hypothesis?
Thank you for your correct remark, although this was only a hypothesis, and the numbers regarding the degree of calcification of the lesions between the two groups did not differ significantly statistically, therefore we believe that artificial differentiation of both groups using OCT figures with different degrees of severity of calcified lesions is pointless, and will be mislead the reader.
- English style and language have to be extensively improved. Several mistakes are present; some concepts are not completely clear.
This was corrcted by a native speaker.
- The “2.2. Optical coherence tomography – image attainment and processing” section is not adequate at all. Considering the relevant role of this section about the methodological approach of the research, more clearness is needed.
I do not fully understand the reviewer's comments, besides, this comment is exactly the opposite of what the first reviewer wrote, who said that the paragraph was very well written.
- There are different problems in text editing. Section “2.3. Definitions” and description of Figure 1 show a different format compared with main text.
This was corrected.
- Check the words spelling in Table 1 (hipercholesterolemia, dyalisotherapy, etc).
This was checked and corrected.
- It is more appropriate to use “poorer stent expansion group” rather than “patients from group of patients with poorer stent expansion”.
This was corrected.
- In Tables, the insertion of median ÷ interquartile range is not necessary.
This is contrary to what the first reviewer wrote, in the end, the data were changed as described in the statistical analyzes, namely in the case of normal distributions, they were presented as mean +/- SD, and in the case of abnormal distribution, medians and IQR.
- Check the description in Figure 1. What does “(< 90 ≥ %)” stand for?
This has been changed on ≥90% vs. <90% stent expansion, for better clarificartion
Reviewer 2 Report
The manuscript “Experience with Optical Coherence Tomography Enhanced by Novel Software (Ultreon 1.0TM) – the First One Hundred Cases”, although on a low number of cases, is looking for predictors of stent underexpansion using OCT as a recognized tool.
The authors compare the “processing results, procedural indices, clinical outcomes according to the extent of stent expansion and assess risk factors of poorer stent expansion in patients treated with PCI using OCT” The technical work is high (they assessed each frame of 100 patients, with 2-3 pullbacks per patients) to quantify the atherosclerotic plaques and to define the stents expansion).
The authors describe the method very accurate, the statistical analysis is well done.
The fact that the procedural characteristics are not predictors means the operators are extremely skilled and no differences are among the two groups.
The results are interesting as predictors for poor expansion are finallyrefering only to the creatinine level.
I consider this research and manuscript acceptable for publication, with the condition to answer to several comments:
1. Please, state clearly at the beginning of the discussion section, the originality and the importance of the study.
2. Please, mention the AIC and BIC of the models analysed.
3. Please calculate and mention the statistical power of the study (1-β).
4. Please, mention at the discussion section, why time of balloon inflation was not taken into consideration.
5. Please, explain what type of balloons were used. High pressure balloon (Schwager) was taken into consideration to prepare the culprit lesions (Secco, Di Mario et al., EuroIntervention, 2016)?
6. Please, consider adding the measurement units for creatinine (table).
7. Please, consider only mean±SD or median (IQR) for each variable, not both. When the characteristic in one group is abnormally distributed use median as the central tendency.
8. Please, describe the definition of hypercholesterolemia used for the study.
9. Please, define kidney failure (table 1). What is the degree of the CKD?
10. Please, consider to present the multivariable analysis for all the variables identified as predictors at the univariate logistic regression analysis, not only for creatinine.
11. What is the sensitivity and specificity for creatinine, and what is the cut-off of the predictor, if calculated?
Author Response

(The authors gave the same response as above.)

Round 2
Reviewer 1 Report
I appreciate the correction made by authors. However, albeit what they declared, different fixes were not made.
1. Text editing still shows problems: "2.3 Definitions" and description of Figure 1, in addition to different tables, show a different format compared with main text again.
2. The words spelling in Table 1 (hipercholesterolemia, dyalisotherapy, etc) was not corrected.
3. The description of Figure 1 was not corrected, as instead declared in cover letter.
4. In Table 6, the creatinine's unit of measure is not defined.
Author Response
I appreciate the correction made by authors. However, albeit what they declared, different fixes were not made.
1.Text editing still shows problems: "2.3 Definitions" and description of Figure 1, in addition to different tables, show a different format compared with main text again.
Tables format wolud be corrected by MDPI. Also Figure 1 format would be corrected by MDPI, format is the same as the old one except for font corrections which were made due to suggestions. Definition =>hypercholesteroleamia was corrected, and I honestly do not see othe problems, could you kindly indicate them?
- The words spelling in Table 1 (hipercholesterolemia, dyalisotherapy, etc) was not corrected.
Hypercholesterolaemia and dialysis therapy were corrected.
- The description of Figure 1 was not corrected, as instead declared in cover letter.
I forgot to correct it inadvertently, it is now being corrected.
- In Table 6, the creatinine's unit of measure is not defined.
This has been added.
